# Evolving Role of Coronary Computed Tomography Angiography (CCTA) in Quantifying Atherosclerotic Coronary Artery Disease: A Narrative Review

**DOI:** 10.3390/diseases13100343

**Published:** 2025-10-16

**Authors:** M. A. Manal Smail, Ram B. Singh, Jan Fedacko, Galal Elkilany, Krasimira Hristova, Sarthak Sharma, Ahmed Bathallah, Sherif A. Baathallah, Monika Jankajova, Fabiola Sozzi

**Affiliations:** 1School of Pharmacy and Biomedical Sciences, University of Central Lancashire, Preston PR1 2HE, UK; farawla-9@hotmail.com; 2Halberg Hospital and Research Institute, Moradabad 244001, India; rbs@tsimtsoum.net; 3University Research Park, PJ Safaric University, 04001 Kosice, Slovakia; janfedacko@hotmail.com; 4Sentara, Northern Virginia Hospital, Woodbridge, VA 22191, USA; galal.elkilany@gmail.com; 5Department of Cardiovascular Imaging, Center for Cardiovascular Disease, 1309 Sofia, Bulgaria; khristovabg@yahoo.com; 6Himalayan Institute of Medical Sciences, Jolly Grant, Dehradun 248140, India; sarthak2293@gmail.com; 7Life Stem, McLean, VA 2210, USA; ahmedbaathallah2009@gmail.com (A.B.); dubai_pgd_international@yahoo.com (S.A.B.); 8Kardiocentrum Agel Saca and UPJS Kosice, 04011 Kosice, Slovakia; 9Department of Cardiology, Milano Majoree Polyclinica, 20122 Milan, Italy; fabiola_sozzi@yahoo.it

**Keywords:** vascular disease, stroke, coronary atherosclerosis, tomography angiography

## Abstract

Background: There have been 20.5 million deaths due to cardiovascular diseases (CVDs), including atherosclerotic coronary artery disease (CAD) and stroke, so far in 2025. Atherosclerosis, which begins in newborns, may be influenced by preconception factors and continues to develop in adults, requiring a proper assessment of the burden of atherosclerotic plaque, as it is the direct cause of CAD. This review aims to emphasize the role of a staging system proposed by the Lancet Commission for the quantification of atherosclerotic coronary artery disease (ACAD) with an emphasis on preconception risk factors and protective factors, based on coronary computed tomography angiography (CCTA). Methods: It is suggested that the use of CCTA scanning makes it possible to quantify the atherosclerotic plaque burden into four stages. Results: CCTA enables us to see how much plaque has built up, as well as the type of plaque, but not the biochemistry of the plaque, to determine its vulnerability. However, if the plaque is a non-calcified fatty plaque, it is considered to be a strong predictor of the risk of myocardial infarction (MI), whereas a more stable calcified plaque is known to be protective against MI. There are several risk factors and protective factors which may influence the process of the rupture or vulnerability of the plaque. A randomized trial revealed that, after a median follow-up of 10·0 years, deaths due to CAD or non-fatal MI were less frequent in the CCTA group compared with a control group. Conclusions: Despite a few gaps in knowledge about the value of a staging system of ACAD, the available evidence indicates that it is helpful in decreasing morbidity and mortality with available therapies.

## 1. Introduction

The Bogalusa Heart Study found that infants dying due to non-cardiac causes had fatty streaks in their coronary arteries in an autopsy study. The study was initially started in 1972 to discover the risk factors of cardiovascular diseases (CVDs) in children and adolescents, which later evolved into a follow-up in young adults [1]. This study demonstrated that atherosclerosis begins early in life, progresses with age, and causes a distribution of lesions in all of the arteries, including coronary arteries. These lesions can predispose individuals to inflammation and athero-thrombosis in later adult life, resulting in coronary artery disease (CAD), acute coronary syndromes, and sudden cardiac death (SCD) [1]. It is important to know the risk factors of CAD and prognosis characterized by the disease burden, impact on quality of life, and average life expectancy [2]. Furthermore, the progression of the diseases and plans for therapy such as drug therapy or surgical intervention, depending upon the stage of the disease, are also crucial [3]. It has been projected that there will be a 90% increase in the prevalence of cardiovascular diseases (CVDs) between 2025 and 2050, with a 73% increase in crude mortality and a 54% increase in crude DALYs [4]. Atherosclerosis begins in newborns and continues to develop in adults [1], requiring a proper assessment of the disease burden of atherosclerotic plaque, as it is the direct cause of atherosclerotic CAD (ACAD). A staging system has also been proposed for the quantification of ACAD [3] based on coronary computed tomography angiography (CCTA) because it is noninvasive, and CAD is a leading cause of death world-wide [2,4]. CAD may be defined as an epicardial coronary artery obstruction resulting in myocardial ischemia due to a reduced supply of oxygen, which may result in increased cardiovascular events and deaths. ACAD is defined as the plaque burden, with progressive atherosclerotic plaques due to the underlying atherosclerosis. The early identification and modification of the underlying atherosclerosis progression (rather than only finding and treating flow-limiting disease or attenuating high-risk characteristics) might eliminate future events.

Recently, the Lancet Commission has published a large review [5] along with comments [6] and an editorial [7] to highlight the importance of CCTA in the early diagnosis of ACAD. Although the risk factors of ACAD have been discussed, there are gaps in our knowledge and in the review about the role of preconception factors in the pathogenesis of ACAD [8,9]. There is no proper emphasis on the role of behavioral risk factors and protective factors, responsible for platelet aggregation and thrombosis [10] in the development of ACAD. There is an emphasis on the biology of atheroma as an important determinant of ACAD, without consideration of the biochemistry of the plaque, platelet function, comorbidities, and role of rupture and erosion, which may be crucial in determining the vulnerability (MMPs) of the plaque [10,11,12,13,14]. In view of the gravity of CVDs, and increased availability of CCTA to diagnose the plaque volume severity, this narrative review aims to re-emphasize the role of the four-stage system proposed for the early diagnosis of ACAD. The novelty of the review is that it also emphasizes the role of preconception factors and protective factors in the pathogenesis and prevention of ACAD.

In this narrative review, we have included the articles concerned with ACAD by conducting a search on Google Scholar, MEDLINE, PubMed, Scopus, and the Cochran database.

## 2. The Four Stages of Atherosclerotic Coronary Artery Disease (ACAD)

There are major biological alterations in the pathogenesis of atherosclerosis, which have been demonstrated to have six stages, in their order of occurrence [13]. These are as follows: 1. the early phase of fatty streaks; 2. the early phase of fibro-atheroma; 3. the thin-cap fibro-atheroma; 4. plaque rupture; 5. the development of the necrotic core; and 6. plaque erosion [13]. These are based on a range of factors that determine the progression of these diseases [13]. Min et al. suggested a four-stage system for the measurement of the progression of CVD which may be based on the atherosclerotic plaque measured by CCTA [3], without any secondary manifestations of heart disease. The quantification of the burden of CAD and the type of atherosclerotic plaque has been found to be the strongest discriminant of the future risk of major adverse cardiac events (MACEs) [2,5,6]. The burden of atherosclerotic plaque also demonstrated a strong correlation with the severity of the stenosis as well as ischemia. According to Min and coworkers, the staging system describes patients based on either the total volume of the plaque or the volume in percent of the atheroma [3]. It is the proportion of arterial walls occupied by plaque and the stages of the plaque, defined as normal (no plaque), mild, moderate, and severe plaque, along with the methods of treatment, that are given in Table 1.

Min et al. included 303 patients aged 64.4 ± 10.2 years, with 71% males, in which CCTA was carried out before ICA and FFR as a clinical indication proposed by AHA/ACC [3]. Other tests such as quantitative computed tomography (QCT) were carried out to assess the percent atheroma volume (PAV, %) and total plaque volume (TPV, mm^3^) [3]. At the 50% quantity coronary angiography (QCA) stenosis threshold, the quantitative computed tomography (QCT) carried out in patients with non-obstructive CAD found a total plaque volume (TPV) of mean (standard deviation, SD) 436 (±444.9) mm^3^ and mean plaque atheroma volume (PAV) of 9.7 (±8.2)%. For the one-vessel disease, a PAV of mean (SD) 11.7 (±8.0)% and TPV of 549.3 (±408.3) mm^3^ were present. For the two-vessel disease, a PAV of mean (SD) 17.8 (±9.8)% and TPV of mean (SD) 838.9 (±550.7) mm^3^ were noted. Interestingly, for the three-vessel disease/left main disease (LMD) in the left main coronary artery, a PAV of mean (SD) 19.2 (±8.2)% and TPV of mean (SD) 799.9 (±357.4) mm^3^ were observed.

Patients with at least one-vessel ischemia (FFR ≤ 0.8) had a PAV of 15.2 (±9.5)% and TPV of 694.6 (±485.1), whereas non-ischemic patients (FFR > 0.8) had a TPV of 422.9 (±387.9 mm^3^) and a mean PAV of 9.2 (±7.3)%. Finally, there were four clinically distinct stages, based on the definition of plaque stages: thresholds of 0, 250, and 750 mm^3^ and 0, 5, and 15% PAV, in which patients with no, non-obstructive, single-vessel, and multi-vessel disease were optimally distributed. The findings revealed that the burden of atherosclerotic plaque by QCT may be related to the severity of the stenosis, its extent, and ischemia. Thus, Min et al. proposed a staging of the plaque burden of atherosclerotic CAD, given in Table 1, using the above definitions.

It is clear that the use of scanning via CCTA can result in the possibility of the quantification of the burden of atherosclerotic plaque. CCTA is helpful in the examination, to find out the quantity of the plaque, as well as the type of plaque. Is it a non-calcified fatty plaque which is considered to be a strong predictor of the risk of ACS or a more stable calcified plaque known to be protective against MI? Therefore, in the absence of a staging system for CAD, it was stemmed because there was no reliable method to assess and treat the risk of heart disease given the current standard of care for CVDs. There is a staging system for coronary calcium scoring, which is primarily used for the prediction of major, adverse clinical events, although it fails to identify the progression of CAD [3] (Figure 1).

## 3. Utility and Necessity of CCTA in Cardiovascular Care

The ability to measure atherosclerotic plaque using CCTA appears to be an important advancement in addressing the management of CAD. It seems that, now, it is possible to develop a personalized and optimal plan of treatment based on the burden of the ACAD. However, it may vary based on the progression of the stage of the disease, thus emphasizing the role of the staging system for the management of CAD [3,13]. The majority of the guidelines have focused on the advice to treat low-density lipoprotein (LDL) cholesterol and other markers of the CAD [2]. Interestingly, the goal of the staging system is to be able to quantify the stage of the disease progression based on the primary cause of CAD, which is the severity of the plaque. The early-stage CAD may be addressed via a modification of one’s diet and lifestyle changes, and antiplatelet therapy or statin therapy, because the buildup of plaque at that particular stage may not have progressed to the extent that angioplasty is needed. Invasive procedures may be advised in patients with more advanced CAD, using the staging system to address the plaque severity (Figure 2).

It seems that the stratification of the risk of CVDs to guide preventive therapy relies on clinical scores based on the identification of plaque vulnerability and risk factor scores, which have a modest discriminative power [16,17,18,19]. CCTA and the coronary artery calcium score (CACS) have emerged as methods for enhancing the estimation of risk and potentially providing insights for personalized therapy for ACAD [16]. The personalized interventions for atherosclerosis are based on the findings of atheroma via CCTA and the risk factors of CVDs, which is advised by the American College of Cardiology [16]. It is possible that the integration of a CCTA-based evaluation of atherosclerosis, clinical guidelines for practice, and the results from a randomized trial can leverage the burden and progression of atherosclerosis as primary targets for therapy. After defining the severity of the stages of atherosclerosis by CCTA, the overall cardiovascular risk prediction may be improved by CACS, which can be useful in the improvement of the results of therapy with statins during primary prevention [16,17]. Therefore, CCTA holds promise in the identification of patients for treatment with antiplatelet agents. CCTA has been chosen for the guidance of therapies for atherosclerotic plaque and to monitor the therapeutic responses in individual patients through the assessment of the features of the individual plaque, the quantification of the total plaque volume as well as the biological composition of the plaque, and assessing the adipose tissue around the coronary artery [17].

## 4. Sensitivity, Specificity, and Predictive Value

There is evidence that the four-stage system for CAD is a concept designed to categorize the severity of plaque buildup in coronary arteries [15,16,17,18,19]. The specificity, sensitivity, and predictive value of this system are crucial for the understanding of its ability to accurately identify and predict the risk of major adverse cardiac events (MACEs), including unstable angina, MI, or death. Sensitivity refers to the ability of the system to correctly identify patients who will develop a MACE. A high sensitivity indicates that the system would not miss many cases of MACE that need timely intervention. Specificity refers to the ability of the system to correctly identify patients who would not develop a MACE. A high specificity is helpful in avoiding unnecessary interventions or concerns for those who are not at high risk. The predictive value encompasses the probability that a positive result may correctly identify an event, and a negative predictive value is the probability that a negative result correctly identifies the absence of an event. These values help determine the accuracy of the system in predicting actual events. It seems that, without evaluating these metrics, the four-stage system may not be considered a reliable tool for predicting the risk of MACEs. The absence of sensitivity, specificity, and predictive value data limits the clinical utility of the system, indicating that the use of an additional scoring system, Gensini, may be useful [20].

This scoring system increases the sensitivity of the assessment of the severity of atherosclerosis. The Gensini score is commonly used for quantifying the severity of CAD based on angiography [20]. It assigns points based on the degree of stenosis in coronary arteries, with higher scores indicating a greater disease burden. Despite it not being a perfect measure, the Gensini score is considered more informative than simply classifying patients based on double-, triple-, or single-vessel disease. It is possible that the four-stage system for CAD is a promising concept, but its potential benefits remain unproven without its rigorous evaluation in impacting patient care.

## 5. The Use of CCTA to Reduce Antithrombotic Therapy Burden

Recently, in 2025, Castiello et al. reiterated the use of CCTA in reducing the burden of antithrombotic therapy that may be pivotal in patients with atrial fibrillation (AF) with an extremely high risk of bleeding, in whom anticoagulant and antiplatelet therapy may be required in the case of the implantation of a drug-eluting stent (DES) [10]. In this condition, the stratification of CCTA may be pivotal to altering the invasive management in patients with AF but without significant plaques. Using aspirin and a P2Y12 inhibitor as dual antiplatelet therapy (DAPT) may be necessary in all patients undergoing PCI for the prevention of complications due to thrombosis. The majority of the patients undergoing PCI also have a coexisting AF, thus requiring an oral anticoagulant (OAC) for the prevention of stroke due to ischemia or a systemic embolism. However, there is an increased risk of bleeding in association with OAC and DAPT. It seems that dual antithrombotic therapy (DAT) has lesser bleeding events, compared with triple antithrombotic therapy (TAT), which may come at a cost of a higher risk of stent thrombosis. Interestingly, AF patients undergoing PCI represent a special population with many challenges, requiring multiple strategies to decrease the risk of complications due to bleeding [10].

Atherosclerosis Treatment Algorithms are presented for the worsening stages of atherosclerosis for patients with risk factors such as diabetes, lipid disorders, hypertension, obesity, and tobacco intake as observed by Lancet Commission [5]. There is a rapid pace of research which should be focused on other risk factors that may improve efforts to optimize precision for the prevention of CAD.

## 6. Risk Factors and Protective Factors Influencing Atherosclerotic Plaque

There is evidence that ACAD can develop over the course of life, beginning from the preconception period, intra-uterine life, and off-spring [1,4], with the early features of atherosclerosis evident as early as the first decade of life [4]. The risk of ACAD may be increased due to multiple factors and may be broadly categorized as genetic, behavioral, biological, metabolic, environmental, or related to other comorbidities such as diabetes mellitus. It seems that several risk factors for ACAD are well-recognized; however, the emphasis is now moving on to novel, emerging, and undiscovered risk factors, which may require further research across diverse populations and age groups [4,5,10]. These risk factors of ACAD are expected to evolve considerably as diet and lifestyle changes, with the alteration in ecology, environment, and demography with the technological unfolding of advancements. These potential confounding variables such as age, sex, and comorbidities may affect the plaque burden, composition, and prognosis. Therefore, any future validation of the staging system should involve multivariate modeling to isolate the independent predictive power.

All the risk factors and protective factors [21,22] may influence the preconception health of the parents [8,9], and, therefore, the health of the sperm and ovum and resultant fertilized egg of the mother, as well as the overall pregnancy and outcome of the newborn [23,24,25]. In a recent study, paternal preconception modifiable risk factors have been found to predispose adverse pregnancy and offspring outcomes [8], which may be important in the pathogenesis of ACAD. Similarly, the relationships between parental modifiable preconception risks and health behaviors were found to have a significant association with the outcomes in mothers and offspring health [8,9]. There is evidence that newborns of parents with obesity may have an alteration in DNA methylation which may be specific to imprinted genes [23], which may predispose obesity in infants and children, possibly along with fatty streaks in the arteries. An experimental study published in nature [24] has found a correlation in the interaction of the epigenetics link with genetics and the environment and diseases such as atherosclerosis [25]. It is clear that the coexistence of risk factors can predispose ACAD, whereas the coexistence of protective factors may decrease the susceptibility to ACAD. All the behavioral risk factors have been found to enhance the risk of atherosclerosis [26,27,28,29,30], some of which have been published in an excellent review by the Lancet Commission [5]. Except for preconception factors and protective factors, all other risk factors such as age, gender, sedentary behavior, obesity, Western dietary pattern, hypertension [31], and diabetes mellitus [32] constitute an epidemic of risk factors which may start in early life with the exposure of risk in utero. In high-income populations, these risk factors may be particularly relevant and in middle-income countries, these are also becoming highly prevalent. The screening for the risk factors of atherosclerosis should, therefore, begin earlier in life in all countries of the world (Table 2).

There is an urgent need to diagnose patients with vulnerable atherosclerotic plaques before the occurrence of thrombosis causing cardiovascular events [18]. In an in vitro experiment, atherosclerotic plaques from five men, aged 79 years, were included in order to find out the features of the rupture-prone plaque. It may be identified by the use of photon-counting computed tomography (PCCT). The plaque images showed calcium deposits identified from all other features [18]. In addition, the rupture-prone plaque had features discernible from each other, compared with a hemorrhage with a fibrous cap (*p* = 0.017), lipids (*p* = 0.003), and necrosis (*p* = 0.004) and thrombus compared to fibrosis (*p* = 0.048), fibrous cap (*p* = 0.028), lipids (*p* = 0.015), and necrosis (*p* = 0.017). It is clear that clinically available PCCT detects calcification, as well as other features of the rupture-prone carotid plaques. It is possible that imaging via computed tomography has benefits from other innovations [19]. In addition, spatial resolutions, a low dose of radiation, the availability of dual- and multi-energy imaging, and the clinical introduction of photon-counting detector CT (PCD-CT) appear to be a milestone with the potential to alter clinical CT imaging with an increase in its indications.

## 7. Morphology and Characteristics of Plaque

The concordance between CCTA and intracoronary imaging techniques (OCT and IVUS) regarding the plaque morphology, such as coronary artery calcium (CAC), could be useful for having a more comprehensive vision of the diagnostic tools to describe and characterize the plaques [40,41]. It is possible that the combination of the CAC score with IVUS and OCT as a strategy in patients with intermediate risk with a family history of CAD may be useful. In a randomized trial, 179 patients were included in the CAC-score-informed and 186 in the usual care groups [40]. Compared with the usual care control group, the CAC-score-informed group revealed a sustained decline in total cholesterol levels at 3 years, in association with a significant decrease in the pooled cohort equation risk calculation, *p* < 0.001. The plaque progression was higher in the control group compared with the CAC group for total plaque volume; *p* = 0.009), in association with a decrease in atherogenic lipids. The REVEALPLAQUE study [41] examined the agreement between AI-enabled quantitative coronary plaque analysis (AI-QCPA) and intravascular ultrasound (IVUS) for the assessment of plaque in the coronary artery. This study reported a strong agreement and high accuracy between AI-QCPA and IVUS for the measurement of the minimal luminal area and total volume of the plaque, and classified the plaque using the system of CAD-RADS. This study also found the diagnostic performance of AI-QCPA for the quantitative detection of significant stenosis.

## 8. Cohort Studies on Progression of Atherosclerotic CAD

Cohort studies using imaging methods such as CT angiography (CTA) and other imaging techniques can track the progression of the plaque and the effect of various therapies on it. These studies often categorize plaque by atheroma volume or volume for a comparison over time and across various groups of treatment. In a case study involving 303 patients, the definition of plaque stage thresholds of 0, 250, and 750 mm^3^, and 0, 5, and 15% percent atheroma volume (PAV) indicated four clinically distinct stages in which patients were distributed with no, non-obstructive, single-vessel disease, and multi-vessel disease [3]. It was concluded that the atherosclerotic plaque burden by quantitative computed tomography (QCT) may be related to the severity of stenosis and the extent as well as ischemia. The authors proposed the staging of the burden of CAD atherosclerotic plaque via the definitions as given earlier in Table 1.

Later on, Freemen and co-workers of the ACC presented the algorithms of treatment which personalize therapies based upon the findings of atherosclerosis via CCTA and cardiovascular risk factors [17]. It seems that, via the integration of a CCTA-based evaluation of atherosclerosis, guideline for clinical practice, and evidence from a controlled trial, the management of Atherosclerosis Algorithms may leverage the specific progression and burden of atherosclerosis as primary targets for intervention. The various stages of atherosclerosis severity are defined by CCTA. This was to be followed by Atherosclerosis Treatment Algorithms that are presented for certain stages of atherosclerosis worsening, for patients with risk factors such as diabetes, obesity, dyslipidemias, high BP, and tobacco intake [17]. It is likely that it may encourage a rapid pace of investigations, due to future needs that can cause an improvement in the efforts for CAD prevention. This approach may allow experts to focus their efforts on assessing and managing the risk of adverse events rather than fixing a lesion or two periodically [42,43,44]. This approach is consistent with precision cardiac care which emphasizes atherosclerosis as the target of primary disease for evaluation and therapy.

According to the guidelines, non-invasive imaging offers a method to identify CAD, to improve risk stratification, and to guide patient management [45,46]. Large randomized controlled trials have demonstrated convincing data regarding the use of CCTA for patients with stable CAD [47,48]. The Scottish Computed Tomography of the Heart (SCOT-HEART) recruited 4146 patients with a mean age of 57 years, with 2325 (56·1%) being male, who were randomized to receive standard care along with CCTA, and 2073 patients who were provided only standard care [47,48]. This controlled study reported that, in patients with suspected CAD having angina, CCTA was associated with a change in diagnosis in 27% of patients, a change in tests in 15%, and an alteration in therapy in 23% [47]. After a follow-up of 5 years, this therapy was associated with a decline in the composite endpoint of CAD death or non-fatal MI as compared with care via standard methods. There were similar rates of coronary interventions, such as PCI and CABG, with a rise in the use of drug treatment with antiplatelet agents and statins, etc. [48].

After a median follow-up of 10·0 years, deaths due to CAD or non-fatal MI were less frequent in the CCTA group compared with the standard care group (137 [6.6%] vs. 171 [8.2%]; hazard ratio [HR] 0·79, *p* = 0·044) [37,49]. While all-cause, cardiovascular, and death due to CAD and stroke showed no difference between the groups (*p* > 0·05 for all), non-fatal MI (90 [4.3%] vs. 124 [6.0%]; hazard ratio (HR) 0·72, *p* = 0·017) and MACE (172 [8.3%] vs. 214 [10.3%]; HR 0·80, *p* = 0·026) were significantly lower in the CCTA group [49]. However, treatment for the prevention of disease was significantly more common in the CCTA group (831 [55.9%] of 1486 vs. 728 [49.0%] of 1485 patients with available data; odds ratio 1·17, *p* = 0·034). It is clear that CCTA-guided therapy, after 10 years, was associated with a significant decline in deaths due to CAD, including non-fatal MI. It is possible that the diagnosis of coronary atherosclerosis by CCTA causes a significant improvement in the long-term prevention of CVDs in patients with stable chest pain with similar rates of coronary revascularisation procedures (15.2% vs. 15.3%, *p* = 0·99). The improved outcomes were due to better drug therapy.

In addition, the protective effects of a more accurate diagnosis may occur due to modifications in diet and lifestyle, the improvement in drug therapy, and the greater use of health services and hospitals. The effects of the treatment with statins continued for more than 16 years in the Scandinavian Legacy study [50], and more than 20 years in the Scotland Study [51]. Interestingly, CAD is a progressive process, which may have attenuated the legacy benefits on outcomes in clinical trials such as the SCOT-HEART trial over the longer-term follow-up. This is found more so in those intervention trials, in which the standard of care includes a greater prescription of therapies aimed at prevention as patients age. There is a need to ascertain if the protective effects of management guided by CCTA persist after 10 years and whether there are subgroups of patients who obtain the maximum benefit.

## 9. Selection of Patients for CCTA and No Indications

CCTA is a non-invasive imaging technique that helps in the diagnosis, and it is typically used in patients with symptoms suggestive of or at a high risk of CAD. The patients selected may already be at a moderate to high risk of CVDs, introducing potential selection bias when generalizing the utility of the staging system to broader populations. This means the study results might not be generalizable to the broader population due to the specific characteristics of those being tested. There is evidence that, in patients with extensive CAC, CCTA had a negative predictive value (>90%) in diagnosing the lack of significant stenosis, and often showed an overestimated presence of coronary stenosis [52]. Therefore, the prevalence of CAD in patients undergoing CCTA is often high, causing a skewed sample, which means the findings may not accurately reflect the true prevalence of CAD in the broader population. The patients that are studied are already a pre-selected group with a higher likelihood of having the condition. This potential bias is crucial for the interpretation of the findings of the studies via CCTA, because a high sensitivity and net percent value in a high-risk population may not translate to the same extent of accuracy in a lower-risk population. It is emphasized that, in borderline cases, shared decision-making is crucial.

## 10. Artificial Intelligence

In patients with a lower risk of CAD, where false-positive results could lead to unnecessary invasive procedures, CCTA may not be the preferred diagnostic method. The integration of artificial intelligence or machine-learning models to refine the staging and outcome prediction could be helpful in this clinical scenario [42]. There is a need to develop high-quality evidence to make this effort meaningful, by using artificial intelligence that has allowed for a more effective automated plaque evaluation across the entire coronary tree [41], and for the prediction of future CVD events [42]. In a case study, involving 351 patients, with a mean age of 65.9 years, the median interval from CCTA to an ACS event was 375 days, and 223 patients (63.5%) had MI. In the derivation cohort *(n* = 243), the best AI-QCPHA features were the fractional flow reserve for the identification of the burden of plaque, lesion, total volume of plaque, low-attenuation volume of plaque, and total myocardial blood flow. There was a higher predictability than the reference model in the validation cohort (n = 108) (AUC: 0.84 vs. 0.78; *p* < 0.001) due to the addition of AI-QCPHA features. It is clear that AI-enabled plaque and hemodynamic quantification enhanced the predictability for culprit lesions in ACS patients, over the conventional coronary CTA analysis.

## 11. Discussion

It is very logical and timely that the importance of the management of atherosclerosis should be focused on individuals on a personal level in patients with ACAD [7]. The message is that it may be important to support the individual and population to commit to actions such as beginning prevention from the preconception period. This approach can help parents to improve the health of the sperm and ovum, when mothers and fathers are planning pregnancy, for a healthy newborn [8,9,10,20,21,22,23,24,25]. This approach may also promote cardiovascular health among newborns, children, and, later on, adults with the aim of the primordial prevention of ACAD. The proposal of the Lancet Commission should be adapted in various health issues, in lower- and middle-income countries, such as those in Africa and Southeast Asia, which have the high burden of CVDs that is rapidly rising.

Progress in reducing the CAD death rate has stalled, although it continues to be a major cause of mortality in the world, with a high disease burden. The Lancet Commission as well as other experts have advised that the diagnosis of ischemia is often made late in patients with atherosclerotic CAD [6,7,8]. The norm should be to manage the whole life-course of atherosclerosis, if the aim is to improve patient outcomes with the objective of the primordial prevention of coronary atherosclerosis. The approaches should be redefined for better cardiovascular health, emphasizing the major challenges in the health systems, such as in medical research and the translation into medical practice [6]. The challenges are how to implement this system, train the experts, decrease the disparities in access by reimbursement, and standardize the approach for the management of ACAD. It seems that a complete shift in planning with the goal of the management of ACAD can save up to 8.7 million lives annually [6,7,8]. It is expected that the saving of lives would be even greater if the aim of the prevention of ACAD begins from the preconception period.

## 12. Early Diagnosis of ACAD

There is a need for a greater emphasis on the early identification of ACAD to find out early plaque or better fatty streaks, to begin diet and lifestyle modification for the prevention of cardiovascular events, which is crucial in the treatment and prevention of CAD. It could be considered an advanced method for preventing myocardial ischemia, a manifestation of late-stage atherosclerosis. There is already so much research for the early identification and treatment of earlier stages of atherosclerosis, providing future protection from CVDs [7,40]. Many cardiologists may examine patients for ACAD via different methods, depending on how rapidly and cost-effectively methods are being developed to image and identify the early stages of atheroma. Cardiologists may possibly choose other methods for the detection of atheroma with predictive algorithms to directly visualize it and quantify its risk for causing events.

In the present state, the strategies for the examination of suspected CAD depend on detecting flow-limiting CAD by testing for functional ischemia or finding out arterial stenosis. The detection of ACAD in the early stages via the use of CCTA should be emphasized, which may alter the end goal of examination for the diagnosis of the future risk of CVDs. This differs from the present method for the detection of coronary obstruction alone in CAD. It is advised that physicians should aim to address ACAD as a longitudinal process, beginning from the preconception period to infancy to childhood, and, later on, among adults and elderly. Atheroma detection may not be a solitary event, but a continuum of the progression of the disease over one’s lifetime such as finding a stenotic vessel or a vulnerable or ruptured plaque. The identification of vulnerable plaques and neovascularization of the atheroma as local inflammation markers are crucial for the prevention of ACAD [11,18]. In vulnerable lesions of carotid plaques, the mRNA levels of matrix metalloproteinase (MMP)-2, -7, -9, and -14 were higher, and the protein levels of MMP-2 and -14 were also increased [18]. There was overexpression of the vascular endothelial growth factor [VEGF], and other markers to indicate new vessel formation and calcium deposition in the vulnerable plaques, indicating the role of MMP-2 and -14 in the increasing vulnerability of the plaque.

There may be a change in clinical practice by cardiologists to address when to start and repeat the examination to determine the adverse effects or favorable changes in the plaque parameters in response to therapy [7]. However, if the approach of managing the whole life-course is adopted, the testing approach may also be modified.

There may be societal questions from the suggested approach for the primordial prevention of atheroma due to the reframing of the risk of CAD from that early preconception period or from infancy. There may be multiple challenges due to this shift in the detection of atherosclerosis at an early stage in its natural history. It seems to be a laudable goal, which may bring along multiple challenges of the cost-effective and safe screening of asymptomatic populations with the target being young parents planning pregnancy. The quality of testing and investigations may need to be altered if people are to be included in lifelong interventions to prevent, arrest, reverse, or stabilize ACAD, and to detect atheroma early (Table 3).

The present agenda proposed by the Lancet Commission [5] may reinvigorate the approach to CAD, but its approach needs to be modified by beginning this process from the preconception period. This may be more common in the Western countries, in populations where individuals who get married at a later age after 40 years when pregnancy may be more often associated with CAD. This could be an important alteration from identifying the luminal compromise and blood flow limitation in the arterial lumen, to the new approach that includes the pathology of the disease in the vascular wall. Several gaps in knowledge still coexist regarding the clinically useful thresholds to understand the plaque burden of the patient for a perfect total guidance on the identification of CAD and its management.

Previous studies found that the remediation of severe stenosis alone could not decrease the major cardiac events among these patients or those with significant ischemia [53,54]. The prognosis of the atherosclerotic plaque and the grade of stenosis may indicate the plaque burden [55], features of vulnerability of plaque [56], and progressive plaque tendency [57] may be crucial for the prediction of the prognosis in CAD. Interestingly, compared with flow-limiting stenosis, non-stenotic plaques are commonly observed [58], which may have a greater risk. Cohort studies provide a further proof that the modification of the burden of plaque or the restriction of the progression of the plaque, or the attenuation of high-risk characteristics can possibly cause the elimination of future CVDs [59]. If, and only if, early diagnosis and treatment is the major determinant of MACEs in patients with chest pain [60], then why not classify ACAD into six stages such as the stages in heart failure [61], which may be helpful in achieving the primordial prevention of ACAD initially via lifestyle modifications? In a recent study, 1041 images were examined [62], showing that AI accurately ruled out obstructive CAD at CCTA and achieved an acceptable agreement with human experts for CAD-RADS 2.0 indicating the severity of the stenosis and plaque burden. Finally, greater emphasis is needed in terms of implementation, challenges, training, standardization, access disparities, and reimbursement as suggested by other experts [5,62], because of the variations in risk factors [63,64,65].

## 13. Conclusions

This communication re-emphasizes the precision cardiac care approach proposed by Lancet Commission, which emphasizes atherosclerosis as the primary target of disease, for evaluation and treatment. Coronary artery atherosclerosis is a significant risk factor for CAD; however, not all patients with atherosclerosis develop CAD, which is clinically significant. CAD is mostly associated with clinical manifestations, such as angina or MI, which may be due to the narrowing of the coronary arteries, not necessarily with plaque. The new approach advocates the use of the coronary atherosclerosis burden and progression to personalize the selection of therapy as well as changes in therapy. The quantification of the burden of ACAD and the type of atherosclerotic plaque by CCTA has been found to be the strongest determinant of the future risk of MACEs; hence, it should be translated into clinical practice with certain economic limitations. There are still gaps in the knowledge on clinically useful thresholds to understand the plaque burden of the patient that need validation for a perfect total guidance on the identification of ACAD. Atherosclerosis is the primary cause of CAD, but the severity of the plaque buildup and its effect on blood flow determine whether CAD develops, and this depends on other factors. In addition, plaque buildup may also be under the influence of behavioral risk factors such as tobacco intake and genetic predisposition, as well as emotional and sleep disorders, which may influence the molecular mechanisms of ACAD, and, therefore, needs to be prevented in clinical practice. The primary reference for the staging system is based on a cohort of only 303 patients; hence, conclusions about the population-level applicability of CCTA may have economic limitations. However, treatment guidance can be made by CCTA in clinical practice because the data provided by SCOT-HEART are strong.

## Figures and Tables

**Figure 1 diseases-13-00343-f001:**
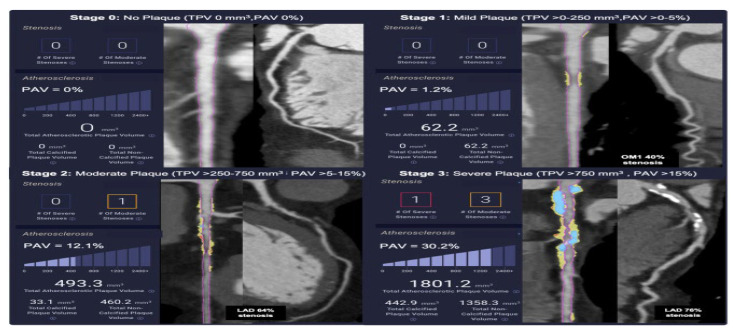
Stages of plaque in clinical examples of plaque disease (adapted from Min et al., reference [3]), under Critical Common attribution.

**Figure 2 diseases-13-00343-f002:**
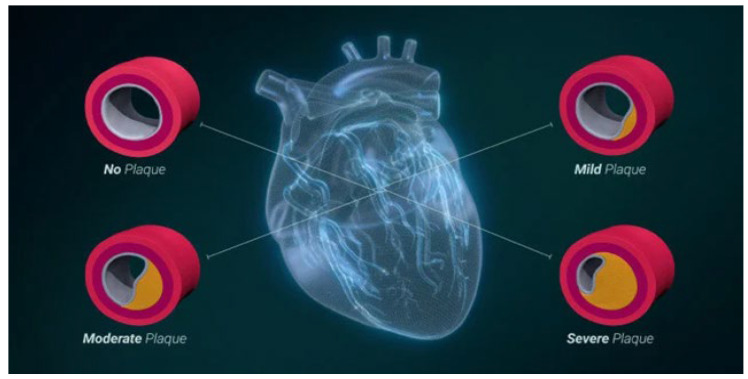
The staging system of coronary artery atherosclerosis (adapted from Min JK, reference [3,15]) under Critical Common attribution.

**Table 1 diseases-13-00343-t001:** Stages of the atherosclerotic plaque and possible approaches for treatment.

Stages of Plaque	Total Plaque Volume (mm^3^)	Percent Atheroma Volume (%)	Medical Therapy
Stage 0: absence of plaque	0, normal	0, normal	Lifestyle modification
Stage 1: mild plaque	>0–250 mm^3^	>0–5%	Guideline-directed; statins
Stage 2: moderate plaque	>250–750 mm^3^	>5–15%	Moderately intensive; high-intensity statins and other agents + bempedoic acid
Stage 3: severe plaque	>750 mm^3^	>15%	Most intensive, high-intensity statins + ezetimibe + PCSK-9i, inclisiran, etc. + bempedoic acid, etc.

Modified from Min JK. New staging system for coronary artery disease. https://cleerlyhealth.com/blog/new-staging-system-for-coronary-artery-disease (accessed on 10 September 2025). Lifestyle modification to be given in all the stages, not advised by Min et al.

**Table 2 diseases-13-00343-t002:** Possible behavioral determinants of health, susceptibility, morbidity, and mortality.

Behavioral Risk Factors	Biological Risk Factors	Protective Factors
Western diet [21]	Hypertension [31]	Mediterranean-style diet [21]
Mental disorders [26]	Diabetes mellitus [32]	Meditation, yoga, and prayer [22]
Sleep disorders [27]	Coronary artery disease [33]	Optimal sleep [27]
Tobacco use [28]	Metabolic syndrome [34]	Moderate physical activity [35]
Alcoholism [29]	Aging [36]	Religious service attendance [37]
Sedentary behavior [30]	Inflammation [38]	Well-being [39]

**Table 3 diseases-13-00343-t003:** Consideration of coronary artery diseases from ischemia to atheroma (modified from Chandrshekhar et al., reference [6]).

Coronary Artery Disease		Atherosclerotic Coronary Artery Disease	
Detection of obstruction	Find coronary stenosis; causes ischemia	Detection of plaque	Characterize coronary plaque, as it is central disease marker
Test for ischemia	Find ischemia; causes events	Stabilize disease	Treat plaque, stop progression, reverse disease; high risk—more adverse events
PCI/CABG	Treat stenosis; reduce ischemia	Revascularize	Revascularize severe stenosis, significant ischemia, symptoms despite OMT-symptom relief
Change, customize	Improvement in symptoms; less consistent benefit in hard endpoints	Change, customize	Finding and modifying plaque, likely to reduce deaths and MI, reduction in need for future revascularization

MI = myocardial infarction, OMT = Optimal Medical Treatment, PCI = percutaneous coronary intervention, CABG = coronary artery bypass grafting.

## Data Availability

No new data were created or analyzed in this study.

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
