# Peer review of "Evolving Role of Coronary Computed Tomography Angiography (CCTA) in Quantifying Atherosclerotic Coronary Artery Disease: A Narrative Review"

_diseases, 2025, doi:10.3390/diseases13100343_

Round 1

Reviewer 1 Report (New Reviewer)

Comments and Suggestions for Authors

Review Summary:

The manuscript addresses a timely topic—the evolving role of CCTA in quantifying atherosclerotic disease burden and guiding therapy through plaque-burden staging. It compiles influential work (e.g., the Lancet Commission perspective and the Min et al. staging framework). It addresses contemporary themes (AI-enabled plaque quantification, photon-counting CT, and long-term outcomes from SCOT-HEART).

However, the paper in its present form reads as a broad position piece with tangential diversions rather than a tightly argued, methodologically transparent narrative review. The principal weaknesses are (i) insufficient description of the review methods (search, selection, synthesis), (ii) uneven depth and balance (e.g., strong emphasis on preconception/epigenetic factors with limited tie-back to CCTA practice), (iii) some technical oversimplifications and unreferenced claims around staging-linked treatment intensity, and (iv) multiple editorial/formatting and figure-rights issues. These shortcomings are correctable, and if addressed, could result in a helpful, practice-relevant narrative review.

Comments to Authors:

  1. Title & Abstract:
    1. Can you simplify the title while retaining its clarity and focus on CCTA-based plaque assessment?
    2. The abstract currently lacks a structured format. Authors are recommended to revise it to include Background, Methods, Results, and Conclusions explicitly.
    3. Please provide concrete, quantitative take-home points (e.g., prognostic accuracy, trial results) rather than general statements.
  2. Introduction:
    1. The introduction repeats epidemiologic data. Authors are advised to streamline and focus more tightly on why imaging plaque burden matters.
    2. The manuscript alternates between CAD and ACAD—can you define these clearly at the start and use one consistently throughout?
    3. Can you better highlight the novelty of this review compared to prior narrative or systematic reviews?
  3. Methods:
    1. At present, the methods are described only as a database search. Can you provide details on:
      1. Databases searched
      2. Date range covered
      3. Keywords/search terms
      4. Language restrictions
      5. Inclusion/exclusion criteria
      6. Study selection process
    2. Even if not a systematic review, could you include a simple flow diagram or table summarizing how evidence was gathered and synthesized?
  4. Staging Framework and Treatment Mapping:
    1. In Table 1, plaque-burden thresholds appear adapted from Min et al, the authors are asked to clarify what is directly adapted versus newly proposed?
    2. Can you separate plaque-burden staging from therapeutic escalation, and support the latter with current guidelines or consensus statements?
    3. Some drug names are misspelled or inconsistently presented (e.g., ezetimibe)
  5. Diagnostic Performance and Limitations:
    1. Authors are advised to include a concise evidence table comparing what CCTA performs well in (e.g., prognostic enrichment, plaque characterization) versus known limitations (e.g., high CAC, radiation, inter-vendor variability)
    2. Statements about the Gensini score “increasing sensitivity” require citations
    3. Include validation studies of the four-stage system’s diagnostic accuracy exist, and summarize their findings
  6. Clinical Trials and Outcomes:
    1. SCOT-HEART data are summarized, but numerical values must exactly match published trials.
    2. Emphasize that improved outcomes were primarily due to better prevention (statins, antiplatelets), rather than differences in revascularization rates
  7. Patient Selection and Non-Indications:
    1. The current section is useful—but can the authors expand with cost-effectiveness considerations, downstream testing burdens, and guideline-based patient suitability?
    2. Authors are advised to emphasize shared decision-making, especially in borderline cases.
  8. Discussion:
    1. Authors are highly advised to expand on implementation challenges: training, standardization, access disparities, and reimbursement
    2. The preconception/epigenetic material is interesting but dominates. Can the authors condense this into a single paragraph that relates to the imaging focus?
  9. Conclusion:
    1. Current conclusions are assertive. Authors are asked to revise them to distinguish between what is ready for clinical practice and what still requires validation.
    2. It would be beneficial to readers if the authors could provide what clinicians can do now versus what remains research territory.

Author Response

Reviewer 2 Report (New Reviewer)

Comments and Suggestions for Authors

Authors used a narrative review to “emphasize the role of a staging system proposed by the Lancet Commission for quantification of the atherosclerotic coronary artery disease (ACAD), based on coronary computed tomography angiography (CCTA)” (L24). However, this article has not fully answered some of the questions due to insufficient description.

First, authors used “±” (e.g., L100), they did not explain what this symbol represents. (i.e., SD or SE). Without explanations, it is difficult for readers to understand what authors did. Authors should add the explanations for their descriptions.

Second, authors showed potential risk factors in Table 2, but they did not explain why they could conclude so (i.e., references as evidence for each risk factor). Without explanations, it is difficult for readers to judge what authors did. Authors should add citations of references as evidence for each risk factor in Table 2.

Finally, authors described some of sentences without citation or justification as follows; “The staging systems of the diseases, are well known for obesity, hypertension, heart failure (HF), type 1 diabetes, type 2 diabetes mellitus (T2DM), and bronchial asthma, based on a range of factors that determine progression of these diseases. There is a need to have a four-stage system for measurement of the progression of CVD which may be based on atherosclerotic plaque in the arteries, without any secondary manifestations of heart disease.” (L76), “Majority of the guidelines have been advising to treat low density lipoprotein (LDL) cholesterol and other markers of the CAD” (L131), “A high sensitivity indicates that the system would not miss many cases of MACE, that need timely intervention. The specificity refers to the ability of the system to correctly identify patients who would not develop a MACE. High specificity is helpful in avoiding unnecessary interventions or concerns for those who are not at high risk.” (L167), “; Gensini may be useful.” (L179), “Using aspirin and a P2Y12 inhibitor, as dual antiplatelet therapy (DAPT) may be necessary in all patients undergoing PCI for prevention of complications due to thrombosis. The majority of the patients undergoing PCI also have coexisting AF, thus requiring an oral anticoagulant (OAC) for prevention of stroke due to ischemia or systemic embolism. However, there is increased risk of bleeding in association of OAC and DAPT.” (L193), “AF patients undergoing PCI represent a special population with many challenges, requiring multiple strategies to decrease the risk of complications due to bleeding.” (L200), “Atherosclerosis Treatment Algorithms are presented for worsening stages of atherosclerosis for patients with risk factors such as, diabetes, lipid disorders, hypertension, obesity, and tobacco intake.” (L203), “The risk of ACAD may be increased due to multiple factors and may be broadly categorized as genetic, behavioral, biological, metabolic, environmental, or related to other comorbidities such as diabetes mellitus.” (L211), “These risk factors of ACAD are expected to evolve considerably as diet and lifestyle change with alteration in ecology, environment, and demography with technological unfolding of advancements.” (L216), “Except for preconception factors and protective factors, all other risk factors such as age, gender, sedentary behavior, obesity, western dietary pattern, diabetes mellitus, and hypertension, constitute an epidemic of risk factors which may start in early life with exposure of risk in utero. In high income populations, these risk factors may be particularly relevant and in middle-income countries, these are also becoming highly prevalent.” (L239), “In an in-vitro experiment, atherosclerotic plaques from 5 men, aged 79 years, were included to find out the features of the rupture-prone plaque.” (L247), “CAD is a progressive process, which may have attenuate the legacy benefits on outcomes in the clinical trials such as SCOT-HEART trial over the longer term follow up. This is more so in those intervention trials, in which the standard of care includes greater prescription of therapies aimed at prevention as patients age. There is a need to ascertain if the protective effects of management guided by CCTA persist after 10 years and whether there are subgroups of patients who get maximum benefit.” (L337), “CCTA is a non-invasive imaging technique that helps the diagnosis and typically used in patients with symptoms suggestive of or on high risk of CAD. The patients selected may already be at moderate to high risk of CVDs, introducing potential selection bias when generalizing the utility of the staging system to broader populations.” (L345), “The patients that are studied are already a pre-selected group with a higher likelihood of having the condition.” (L355), “Progress in decreasing mortality due to CAD has stalled, although, CAD remains a leading cause of death worldwide with a large source of disease burden.” (L388), “The whole life-course management of atherosclerosis should be the norm if patient outcomes are to be improved and the aim is primordial prevention of ACAD.” (L), and “The primary reference for the staging system is based on a cohort of only 303 patients” (L479), but it is difficult for readers to judge them without references as evidence for each description. Authors should add references for these descriptions.

Minor comments

L9: “India, "Ram B. Singh" <rbs@tsimtsoum.net>” may be typo of “India, rbs@tsimtsoum.net”.

L11: “VA, USA. <galal.elkilany@gmail.com>” may be typo of “VA, USA. galal.elkilany@gmail.com”.

L12: “Disease, , Sofia, Bulgaria, <khristovabg@yahoo.com> ” may be typo of “Disease, Sofia, Bulgaria, khristo-vabg@yahoo.com”.

L14: “<sarthak2293@gmail.com>” may be typo of “sarthak2293@gmail.com”.

L92: “2025.Life-” may be typo of “2025. Life-”.

L227: “Table 2.” may be typo.

L626: “Jankajova M, Fedacko J, Singh RB, Fatima G. Preconception Risk factors and Protective Factors of Atherosclerotic Coronary Artery Disease (ACAD). Lancet 2025; 405: in press” may be typo.

Author Response

Authors used a narrative review to “emphasize the role of a staging system proposed by the Lancet Commission for quantification of the atherosclerotic coronary artery disease (ACAD), based on coronary computed tomography angiography (CCTA)” (L24). However, this article has not fully answered some of the questions due to insufficient description.

1.First, authors used “±” (e.g., L100), they did not explain what this symbol represents. (i.e., SD or SE). Without explanations, it is difficult for readers to understand what authors did. Authors should add the explanations for their descriptions. OK, Response: Thank you, corrected standard deviation

2.Second, authors showed potential risk factors in Table 2, but they did not explain why they could conclude so (i.e., references as evidence for each risk factor). Without explanations, it is difficult for readers to judge what authors did. Authors should add citations of references as evidence for each risk factor in Table 2.

Response: 11,References. Given for each,

Table 2. Possible behavioral determinants of health, susceptibility, morbidity and mortality.

Behavioral risk factors

Biological risk factors

Protective factors

         Western diet[21]

Hypertension[31]

Mediterranean-style diet [21]

         Mental disorders[26]

Diabetes mellitus[32]

Moderate physical activity[37]

         Sleep disorders [27]

Coronary artery disease[33]

Religious service attendance[38]

Tobacco use[28]

Metabolic syndrome[34]

Wellbeing [39]

Alcoholism[29]

Aging[35]

Optimal sleep [27]

Sedentary behavior[30]

Inflammation[36]

Meditation, yoga and prayer[22]

References

There are major pathological alterations in the pathogenesis of atherosclerosis which have been demonstrated to have six stages, in their order of occurrence. These are;1. early phase of fatty streak,2. early phase of fibro-atheroma,3. thin-cap fibro-atheroma, 4.plaque rupture, 5.development of the necrotic core and 6. plaque erosion

21.Abusheikha AJ, Johnson CSC, Snyder-Mackler N, Zimmerman KD, Negrey JD, Chiou KL, Frye BM, Howard TD, Shively CA, Register TC. Differential effects of Mediterranean vs. Western diets on coronary atherosclerosis and peripheral artery transcriptomics. Front Nutr. 2025 Jul 10;12:1564741. doi: 10.3389/fnut.2025.1564741.

22.Levine GN,   Lange RA, C. Noel Bairey‐Merz CN.  Meditation and Cardiovascular Risk Reduction: A Scientific Statement From the American Heart Association. JAHA 2017;6: https://doi.org/10.1161/JAHA.117.00221

29.Kiechl S, Willeit J, Rungger G, Egger G, Oberhollenzer F, Bonora E. Alcohol consumption and atherosclerosis: what is the relation? Prospective results from the Bruneck Study. Stroke. 1998 May;29(5):900-7. doi: 10.1161/ 01.str.29.5.900.

30.Shiraishi N, Suzuki Y, Kuromatsu I, Komiya H, Kuzuya M. Sedentary behavior is associated with arteriosclerosis in frail older adults. Nagoya J Med Sci. 2022 Feb;84(1):91-100. doi: 10.18999/nagjms.84.1.91.

31.Poznyak AV, Sadykhov NK, Kartuesov AG, Borisov EE, Melnichenko AA, Grechko AV, Orekhov AN. Hypertension as a risk factor for atherosclerosis: Cardiovascular risk assessment. Front Cardiovasc Med. 2022 Aug 22;9:959285. doi: 10.3389/fcvm.2022.959285.

32.Ye J, Li L, Wang M, Ma Q, Tian Y, Zhang Q, Liu J, Li B, Zhang B, Liu H, Sun G. Diabetes Mellitus Promotes the Development of Atherosclerosis: The Role of NLRP3. Front Immunol. 2022 Jun 29;13:900254. doi: 10.3389/fimmu.2022.900254.

33.Achim A, Péter OÁ, Cocoi M, Serban A, Mot S, Dadarlat-Pop A, Nemes A, Ruzsa Z. Correlation between Coronary Artery Disease with Other Arterial Systems: Similar, Albeit Separate, Underlying Pathophysiologic Mechanisms. J Cardiovasc Dev Dis. 2023 May 11;10(5):210. doi: 10.3390/jcdd10050210.

34.Fernando Yue Cesena, Metabolic syndrome and premature atherosclerotic cardiovascular disease: insights for the individual and the population, European Journal of Preventive Cardiology, Volume 31, Issue 10, August 2024, Pages 1301–1302, https://doi.org/10.1093/eurjpc/zwae139

35.Wong, J.J., Hong, R., Teo, L.L.Y. et al. Atherosclerotic cardiovascular disease in aging and the role of advanced cardiovascular imaging. npj Cardiovasc Health 2024;1, 11 (2024). https://doi.org/10.1038/s44325-024-00012-y

36.Ajoolabady, A., Pratico, D., Lin, L. et al. Inflammation in atherosclerosis: pathophysiology and mechanisms. Cell Death Dis 15, 817 (2024). https://doi.org/10.1038/s41419-024-07166-8

37.Osawa Y, Arai Y. Preventive Effects of Physical Activity on the Development of Atherosclerosis: A Narrative Review. J Atheroscler Thromb. 2025 Jan 1;32(1):11-19. doi: 10.5551/jat.RV22029. Epub 2024 Oct 22

38.Feinstein M, Liu K, Ning H, Fitchett G, Lloyd-Jones DM. Burden of cardiovascular risk factors, subclinical atherosclerosis, and incident cardiovascular events across dimensions of religiosity: The multi-ethnic study of atherosclerosis. Circulation. 2010 Feb 9;121(5):659-66. doi: 10.1161/CIRCULATIONAHA.109.879973.

39.Glenn N. Levine, MD, FAHA, Chair, Beth E. Cohen, MD, MAS, Yvonne Commodore-Mensah, PhD, MHS, RN, Julie Fleury, PhD, Jeff C. Huffman, Psychological Health, Well-Being, and the Mind-Heart-Body Connection: A Scientific Statement From the American Heart Association. Circulation 2021; 143 (10): https://doi.org/10.1161/ CIR. 000000000 000094

Ishida M, Sakai C, Kobayashi Y, Ishida T. Cigarette Smoking and Atherosclerotic Cardiovascular Disease. J Atheroscler Thromb. 2024 Mar 1;31(3):189-200. doi: 10.5551/jat.RV22015.

4.Higueras-Fresnillo S, Herraiz-Adillo Á, Ahlqvist VH, Öberg R, Lenander C, Wennberg P, Wångdahl J, Berglind D, Daka B, Östgren CJ, Rådholm K, Henriksson P. Associations of psychological factors with atherosclerosis and cardiovascular health in middle-age: the population-based Swedish Cardio Pulmonary bioImage study (SCAPIS). BMC Public Health. 2024 May 30;24(1):1455. doi: 10.1186/s12889-024-18924-w.

5.Kadoya M, Koyama H. Sleep, Autonomic Nervous Function and Atherosclerosis. Int J Mol Sci. 2019 Feb 13;20(4):794. doi: 10.3390/ijms20040794.

3.Finally, authors described some of sentences without citation or justification as follows; “The staging systems of the diseases, are well known for obesity, hypertension, heart failure (HF), type 1 diabetes, type 2 diabetes mellitus (T2DM), and bronchial asthma, based on a range of factors that determine progression of these diseases.

Response: This sentence is replaced by new lines with new reference.

13.Pahwa R, Jialal I. Atherosclerosis.https://www.ncbi.nlm.nih.gov/books/NBK507799/, Updated August 2023, Treasure Island (FL): Stat Pearls Publishing; accessed Sept 2025

There are major biological alterations in the pathogenesis of atherosclerosis, which have been demonstrated to have six stages, in their order of occurrence [13]. These are;1. early phase of fatty streak,2. early phase of fibro-atheroma,3. thin-cap fibro-atheroma, 4. plaque rupture, 5. development of the necrotic core and 6. plaque erosion [13]. These are based on a range of factors that determine progression of these diseases

There is a need to have a four-stage system for measurement of the progression of CVD which may be based on atherosclerotic plaque in the arteries, without any secondary manifestations of heart disease.”(L76), “Majority of the guidelines have been advising to treat low density lipoprotein (LDL) cholesterol and other markers of the CAD” (L131), “A high sensitivity indicates that the system would not miss many cases of MACE, that need timely intervention. The specificity refers to the ability of the system to correctly identify patients who would not develop a MACE. High specificity is helpful in avoiding unnecessary interventions or concerns for those who are not at high risk.” (L167), “; Gensini may be useful.” (L179), “Using aspirin and a P2Y12 inhibitor, as dual antiplatelet therapy (DAPT) may be necessary in all patients undergoing PCI for prevention of complications due to thrombosis. The majority of the patients undergoing PCI also have coexisting AF, thus requiring an oral anticoagulant (OAC) for prevention of stroke due to ischemia or systemic embolism. However, there is increased risk of bleeding in association of OAC and DAPT.” (L193), “AF patients undergoing PCI represent a special population with many challenges, requiring multiple strategies to decrease the risk of complications due to bleeding.” (L200), “Atherosclerosis Treatment Algorithms are presented for worsening stages of atherosclerosis for patients with risk factors such as, diabetes, lipid disorders, hypertension, obesity, and tobacco intake.” (L203), “The risk of ACAD may be increased due to multiple factors and may be broadly categorized as genetic, behavioral, biological, metabolic, environmental, or related to other comorbidities such as diabetes mellitus.” (L211), “These risk factors of ACAD are expected to evolve considerably as diet and lifestyle change with alteration in ecology, environment, and demography with technological unfolding of advancements.” (L216), “Except for preconception factors and protective factors, all other risk factors such as age, gender, sedentary behavior, obesity, western dietary pattern, diabetes mellitus, and hypertension, constitute an epidemic of risk factors which may start in early life with exposure of risk in utero. In high income populations, these risk factors may be particularly relevant and in middle-income countries, these are also becoming highly prevalent.” (L239), “In an in-vitro experiment, atherosclerotic plaques from 5 men, aged 79 years, were included to find out the features of the rupture-prone plaque.” (L247), “CAD is a progressive process, which may have attenuate the legacy benefits on outcomes in the clinical trials such as SCOT-HEART trial over the longer term follow up. This is more so in those intervention trials, in which the standard of care includes greater prescription of therapies aimed at prevention as patients age. There is a need to ascertain if the protective effects of management guided by CCTA persist after 10 years and whether there are subgroups of patients who get maximum benefit.” (L337), “CCTA is a non-invasive imaging technique that helps the diagnosis and typically used in patients with symptoms suggestive of or on high risk of CAD. The patients selected may already be at moderate to high risk of CVDs, introducing potential selection bias when generalizing the utility of the staging system to broader populations.” (L345), “The patients that are studied are already a pre-selected group with a higher likelihood of having the condition.” (L355), “Progress in decreasing mortality due to CAD has stalled, although, CAD remains a leading cause of death worldwide with a large source of disease burden.” (L388), “The whole life-course management of atherosclerosis should be the norm if patient outcomes are to be improved and the aim is primordial prevention of ACAD.” (L), and “The primary reference for the staging system is based on a cohort of only 303 patients” (L479), but it is difficult for readers to judge them without references as evidence for each description. Authors should add references for these descriptions.

Response: Sir, these are well known facts so do not need references, but we are giving references in many of them.

Minor comments: 

L9: “India, "Ram B. Singh" <rbs@tsimtsoum.net>” may be typo of “India, rbs@tsimtsoum.net”.

L11: “VA, USA. <galal.elkilany@gmail.com>” may be typo of “VA, USA. galal.elkilany@gmail.com”.

L12: “Disease, , Sofia, Bulgaria, <khristovabg@yahoo.com> ” may be typo of “Disease, Sofia, Bulgaria, khristo-vabg@yahoo.com”.

L14: “<sarthak2293@gmail.com>” may be typo of “sarthak2293@gmail.com”.

L92: “2025.Life-” may be typo of “2025. Life-”.

L227: “Table 2.” may be typo.

L626: “Jankajova M, Fedacko J, Singh RB, Fatima G. Preconception Risk factors and Protective Factors of Atherosclerotic Coronary Artery Disease (ACAD). Lancet 2025; 405: in press” may be typo. deleted

Response: All above mistakes corrected

Reviewer 3 Report (New Reviewer)

Comments and Suggestions for Authors

The manuscript is well-structured, comprehensively referenced and fills an important gap in current cardiovascular risk stratification. The authors know very well the limitations of the study, such as sample size constraints and the need for validation of the staging system in diverse populations. Some concepts (e.g., preconception factors, life-course approach) are repeated across sections without significant progression. Minor grammatical errors and formatting inconsistencies are observed. I have the impression the manuscript would benefit from a more critical tone. 

Author Response

The manuscript is well-structured, comprehensively referenced and fills an important gap in current cardiovascular risk stratification. The authors know very well the limitations of the study, such as sample size constraints and the need for validation of the staging system in diverse populations. Some concepts (e.g., preconception factors, life-course approach) are repeated across sections without significant progression. Minor grammatical errors and formatting inconsistencies are observed. I have the impression the manuscript would benefit from a more critical tone. 

Response: Thank you for valuable comments

Reviewer 4 Report (New Reviewer)

Comments and Suggestions for Authors

This review is very informative and it discusses the critical role of Coronary Computed Tomography Angiography (CCTA) in evaluating Atherosclerotic Coronary Artery Disease (ACAD), a leading cause of global death. The review is advocating a four-stage system for plaque burden quantification using CCTA, that aims to provide a more precise and personalized approach disease management over traditional methods solely focusing on ischemia. The review explored the adaptability of CCTA in cardiovascular care and disease management with focused assumption in predicting major adverse cardiac events (MACE), and its potential to guide anti-thrombotic therapies. The review also discusses the influence of various risk factors, including preconception factors and lifestyle choices, on ACAD development and progression. The review emphasizes the importance of early diagnosis and a lifestyle management approach to prevent ACAD, suggesting that future research, potentially incorporating artificial intelligence (AI), would further guide diagnostic certainties and improved therapeutic strategies.

  1. A key limitation of the review is the lack of clear data on the impact of staging system on reducing morbidity and mortality. Identification of plaque type by CCTA does not necessarily determine or guides biochemical vulnerability. Rupture prediction by employing CCTA is still a debated area.
  2. The review specifically analyzed data obtained from a case study consisting 303 patients which though not very small but still limited and inclusion of more patient data would probably make the conclusions more generalizable. The CCTA studies also appeared to confounded by selection bias, inclusion of high-risk patients that might potentially affected the generalizability and accuracy of the study.
  3. Constructing a table highlighting the clinical trials (associated CT#, phase, design, outcome data) could be beneficial.
  4. The main conclusion is that atherosclerosis and not just its symptoms must be the primary target for disease management. CCTA's ability to accurately quantify plaque burden has potential for positively predicting MACE. This would guide a whole life-course impacting primordial prevention strategies that focuses mainly on changing sociocultural and environmental conditions. mitigating the disease risk factors in the first place, rather than addressing existing risks.
  5. Multivariate modeling may be helpful for validating the predictive power of the staging system. The integration of Artificial Intelligence (AI) seems crucial for refining staging, automated plaque analysis, and improving outcome predictions.
  6. The Conclusions section may be redefined as ‘Conclusions and future directions’ and that would help accommodate capturing the possible approaches for risk mitigation and lifestyle modifications.

Author Response

This review is very informative and it discusses the critical role of Coronary Computed Tomography Angiography (CCTA) in evaluating Atherosclerotic Coronary Artery Disease (ACAD), a leading cause of global death. The review is advocating a four-stage system for plaque burden quantification using CCTA, that aims to provide a more precise and personalized approach disease management over traditional methods solely focusing on ischemia. The review explored the adaptability of CCTA in cardiovascular care and disease management with focused assumption in predicting major adverse cardiac events (MACE), and its potential to guide anti-thrombotic therapies. The review also discusses the influence of various risk factors, including preconception factors and lifestyle choices, on ACAD development and progression. The review emphasizes the importance of early diagnosis and a lifestyle management approach to prevent ACAD, suggesting that future research, potentially incorporating artificial intelligence (AI), would further guide diagnostic certainties and improved therapeutic strategies. Response: Thank you, valuable comments.

1. A key limitation of the review is the lack of clear data on the impact of staging system on reducing morbidity and mortality. Identification of plaque type by CCTA does not necessarily determine or guides biochemical vulnerability. Rupture prediction by employing CCTA is still a debated area.

Response: Surely, mentioned, SCOT-HEART showed decline in mortality.Ref. 47, 48

2. The review specifically analyzed data obtained from a case study consisting 303 patients which though not very small but still limited and inclusion of more patient data would probably make the conclusions more generalizable. The CCTA studies also appeared to confounded by selection bias, inclusion of high-risk patients that might potentially affected the generalizability and accuracy of the study.

Response: Thank you, you are right

3. Constructing a table highlighting the clinical trials (associated CT#, phase, design, outcome data) could be beneficial.

Response: Not very many trials, SCOT-HEART presented in detail.Ref 48

4. The main conclusion is that atherosclerosis and not just its symptoms must be the primary target for disease management. CCTA's ability to accurately quantify plaque burden has potential for positively predicting MACE. This would guide a whole life-course impacting primordial prevention strategies that focuses mainly on changing sociocultural and environmental conditions. mitigating the disease risk factors in the first place, rather than addressing existing risks.

Response: Surely, Thank you

5. Multivariate modeling may be helpful for validating the predictive power of the staging system. The integration of Artificial Intelligence (AI) seems crucial for refining staging, automated plaque analysis, and improving outcome predictions. Response: Thank you, mentioned

6. The Conclusions section may be redefined as ‘Conclusions and future directions’ and that would help accommodate capturing the possible approaches for risk mitigation and lifestyle modifications.

Response: Modified. New References added

13.Pahwa R, Jialal I. Atherosclerosis. https://www.ncbi.nlm.nih.gov/books/NBK507799/, Updated August 2023, Treasure Island (FL): Stat Pearls Publishing;  accessed Sept 2025

62.Kay FU ,  Canan A,  Kukkar V et al. Diagnostic Accuracy of On-Premise Automated Coronary CT Angiography Analysis Based on Coronary Artery Disease Reporting and Data System 2.0.Radiology, 2025;315(2): https://doi.org/10.1148/radiol.242087

In a recent study, 1041 images were examined [62], showing that AI accurately ruled out obstructive CAD at CCTA and achieved acceptable agreement with human experts for CAD-RADS 2.0 indicating severity of stenosis and plaque burden.

15.Kiss MG, Cohen O, McAlpine CS, Swirski FK. Influence of sleep on physiological systems in atherosclerosis. Nat Cardiovasc Res. 2024 Nov;3(11):1284-1300. doi: 10.1038/s44161-024-00560-7. Epub 2024 Nov 8.

16.Murugesan P. Yoga and Cardiovascular Diseases - A Mechanistic Review. Int J Yoga. 2024 May-Aug;17(2):83-92. doi: 10.4103/ijoy.ijoy_55_24.

Xue, T, Chiao B, Xu T et al. The heart-brain axis: A proteomics study of meditation on the cardiovascular system of Tibetan Monks. eBioMedicine, 2022; 80, 104026

Brewer LPC, Bowie J, Slusser JP. Religiosity/Spirituality and Cardiovascular Health: The American Heart Association Life's Simple 7 in African Americans of the Jackson Heart Study JAHA 2022;11: https:// doi. org/10.1161/JAHA.121.02497

Round 2

Reviewer 2 Report (New Reviewer)

Comments and Suggestions for Authors

Authors revised the manuscript, but this article has not fully answered some of the questions due to insufficient description.

In fact, as mentioned in the previous review, authors used “±”, they did not explain what this symbol represents. (i.e., SD or SE). Authors suggest “Thank you, corrected standard deviation”, but Without explanations in the manuscript, it is difficult for readers to understand what authors did. Authors should add the explanations for their descriptions.

Author Response

The Editor

The data are in mean (standard Deviation,SD) as given in the attached MS. For example total plaque volume of mean (SD) 436+/-444.9

Reviewer 4 Report (New Reviewer)

Comments and Suggestions for Authors

The authors have provided satisfactory responses to all my questions/comments. The revised manuscript has now been much improved.

Author Response

Min et al, included 303 patients aged 64.4 ​± ​10.2 years; 71% males, in which CCTA was done, before ICA and FFR as a clinical indication proposed by AHA/ACC [3]. Other tests such as Quantitative computed tomography (QCT) was done to assess percent atheroma volume (PAV, %) and total plaque volume (TPV, mm3) [3]. At the 50% quantity coronary angiography (QCA) stenosis threshold, quantitative computed tomography (QCT) done in patients with non-obstructive CAD, found a total plaque volume (TPV) of mean (standard deviation, SD) 436 (±444.9) mm3. and mean plaque atheroma volume (PAV) of 9.7 (±8.2) %. For 1 vessel disease, the PAV of, mean (SD) 11.7 (±8.0)% and TPV of 549.3 (±408.3) mm3 were present. For 2 vessel disease, the PAV of mean (SD)  17.8 (±9.8)% and TPV of mean (SD) 838.9 (±550.7) mm3 were noted. Interestingly, for 3 vessel disease /left main disease (LMD) in the left main coronary artery, the PAV of mean (SD)  19.2 (±8.2)% and TPV of mean (SD)  799.9 (±357.4) mm3 were observed.

Round 3

Reviewer 2 Report (New Reviewer)

Comments and Suggestions for Authors

Authors revised the manuscript, and I have no further comments.

This manuscript is a resubmission of an earlier submission. The following is a list of the peer review reports and author responses from that submission.

Round 1

Reviewer 1 Report

Comments and Suggestions for Authors

In this narrative review, Dr. Ram B Singh and colleagues aimed to emphasize the role of a staging system proposed by the Lancet Commission for quantification of the atherosclerotic coronary artery disease (ACAD), based on coronary computed tomography angiography (CCTA).

The authors explored the role of CCTA to classify the CAD severity, proposing management strategies influencing atherosclerotic plaque progression.

The paper research topic warrants careful consideration, due to strong interest about this recent switch from functional-based approach to an anatomical-based approach about CAD management with a deeper focus on the plaque rather than on the ischemia.

Overall, this is a quite nicely written article with good figures and tables.

However, it has some limitations that should be addressed from the authors:

  • Considering that this is a narrative review the authors should specify that in the title of the manuscript. Furthermore, the authors should discuss the source of the included studies in this review (e.g. PubMed, MEDLINE).

  • While the four-stage system is appealing in concept, the manuscript does not evaluate or report its sensitivity, specificity, or predictive value for major adverse cardiac events (MACE). This significantly weakens the claim that it can guide clinical decisions.

  • In the manuscript there is only a small mention about the antiplatelet therapy required based on CCTA findings. However, a paragraph about antithrombotic therapy in patients with CAD is required. The authors should discuss about the strategies to reduce bleeding risk including, away from the use of CCTA to defer invasive strategies if not required, DAPT shortening and P2Y12-inhibitors monotherapy after PCI with DES. In this respect, the authors should cite this relevant paper: doi:10.1016/j.ahj.2022.10.006.

  • Along the previous point, the use of CCTA to reduce antithrombotic therapy burden could be pivotal in atrial fibrillation (AF) patients with extremely high bleeding risk in whom anticoagulant and antiplatelet therapy should be needed in case of DES implantation. In this clinical scenario, the CCTA stratification could be pivotal to defer invasive management in patients with AF but without significant plaques. In this topic the authors could cite this recent paper discussing about the outcomes optimization strategies in AF patients undergoing PCI: doi:10.3390/jcdd12040142.

  • A brief mention of the concordance between CCTA and intracoronary imaging techniques (OCT and IVUS) about the plaque morphology could be useful to have a more comprehensive vision of the diagnostic tools to describe and characterize the plaques.

  • There is no mention of longitudinal data showing how plaque staging progresses over time in individual patients or how medical therapy modifies these stages. Without time-series data, the value of staging as a management tool is limited.

  • The primary reference for the staging system (Min et al.) is based on a cohort of only 303 patients. Conclusions about population-level applicability or treatment guidance should be made cautiously, with acknowledgment of the sample size limitation.

  • The review does not critically discuss the potential confounding variables (e.g., age, sex, comorbidities) that may affect plaque burden, composition, and prognosis. Any future validation of the staging system should involve multivariate modeling to isolate independent predictive power.

  • The review does not address that patients selected for CCTA may already be at moderate to high cardiovascular risk, introducing potential selection bias when generalizing the utility of the staging system to broader populations.

  • In the manuscript it could be helpful adding a brief section discussing the clinical scenarios in which the CCTA could not be the preferred diagnostic tool to find and manage CAD.

  • The integration of artificial intelligence or machine learning models to refine staging and outcome prediction could be helpful in this clinical scenario. Please, expand this section.

  • To improve the impact of this paper the authors could provide a graphical picture with CCTA, plaques and stenting images discussing the key advantages and disadvantages of the discussed staging classification of ACAD.

  • Furthermore, a structured table summarizing the strengths and limitations of the cited studies on CCTA and plaque staging. Could be really appreciable.

  • Please, reconcile the text’s character in all the text and use only an order of numbers in the bibliography.

Author Response

Refree 1.

In this narrative review, Dr. Ram B Singh and colleagues aimed to emphasize the role of a staging system proposed by the Lancet Commission for quantification of the atherosclerotic coronary artery disease (ACAD), based on coronary computed tomography angiography (CCTA). We are grateful to the Referee, who has worked hard and is much more learned than authors of this MS.

The authors explored the role of CCTA to classify the CAD severity, proposing management strategies influencing atherosclerotic plaque progression. The paper research topic warrants careful consideration, due to strong interest about this recent switch from functional-based approach to an anatomical-based approach about CAD management with a deeper focus on the plaque rather than on the ischemia. THANKS

Overall, this is a quite nicely written article with good figures and tables.  However, it has some limitations that should be addressed from the authors: THANKS

  • Considering that this is a narrative review the authors should specify that in the title of the manuscript. Furthermore, the authors should discuss the source of the included studies in this review (e.g. PubMed, MEDLINE).GIVEN, last lines under Introduction

In this narrative review, we have included the articles concerned with ACAD by conducting search of Google Scholar, MEDLINE, PubMed, Scopus and Cochran data base.

2.While the four-stage system is appealing in concept, the manuscript does not evaluate or report its sensitivity, specificity, or predictive value for major adverse cardiac events (MACE). This significantly weakens the claim that it can guide clinical decisions. GIVEN
Sensitivity, Specificity and Predictive Value

There is evidence that the four-stage system for CAD is a concept designed to categorize the severity of plaque buildup in coronary arteries [15-19]. The specificity, sensitivity, and predictive value of this system, are crucial for understanding its ability to accurately identify and predict the risk of major adverse cardiac events (MACE), including; unstable angina, myocardial infarction, or death. The sensitivity refers to the ability of the system to correctly identify patients who will develop a MACE. A high sensitivity indicates that the system would not miss many cases of MACE, that need timely intervention. The specificity refers to the ability of the system to correctly identify patients who would not develop a MACE. High specificity is helpful in avoiding unnecessary interventions or concerns for those who are not at high risk. The predictive value encompasses the probability that a positive result may correctly identify an event and negative predictive value, the probability that a negative result correctly identifies the absence of an event. These values help determine the accuracy of the system in predicting actual events. It seems that without evaluating these metrics, the four-stage system, may not be considered a reliable tool for predicting the risk of MACE. The absence of sensitivity, specificity, and predictive value data limits the clinical utility of the system. It is possible that the four-stage system for CAD is a promising concept, but its potential benefits remain unproven without its rigorous evaluation in impacting patient care. 

3.In the manuscript there is only a small mention about the antiplatelet therapy required based on CCTA findings. However, a paragraph about antithrombotic therapy in patients with CAD is required. The authors should discuss about the strategies to reduce bleeding risk including, away from the use of CCTA to defer invasive strategies if not required, DAPT shortening and P2Y12-inhibitors monotherapy after PCI with DES. In this respect, the authors should cite this relevant paper: doi:10.1016/j.ahj .2022.10.006 .Not Found, following about antithrombotic therapy given.

The Use of CCTA to Reduce Antithrombotic Therapy Burden

Recently, in 2025, Castiello et al, reiterated the use of CCTA in reducing the burden of antithrombotic therapy that may be pivotal in patients with atrial fibrillation (AF) with extremely high risk of bleeding in whom anticoagulant and antiplatelet therapy may be required in case of implantation of drug eluting stent (DES)[10]. In this condition, the stratification of CCTA may be pivotal to alter invasive management in patients with AF but without significant plaques. Using aspirin and a P2Y12 inhibitor, as dual antiplatelet therapy (DAPT) may be necessary in all patients undergoing PCI for prevention of complications due to thrombosis. The majority of the patients undergoing PCI also have coexisting AF, thus requiring an oral anticoagulant (OAC) for prevention of stroke due to ischemia or systemic embolism. However, there is increased risk of bleeding in association of OAC and DAPT. It seems that dual antithrombotic therapy (DAT) has lesser bleeding events, compared with a triple antithrombotic therapy (TAT), that may be at the cost of higher risk of stent thrombosis. Interestingly, AF patients undergoing PCI represent a special population with many challenges, requiring multiple strategies to decrease the risk of complications due to bleeding.

Wang, Z., Zhu, S., Zhu, J. et al. Comparison of P2Y12 inhibitors and aspirin in secondary prevention of coronary events: a meta-analysis of RCTs. BMC Cardiovasc Disord 25, 207 (2025). https://doi.org/10.1186/s12872-025-04668-x

Anastasios Apostolos, Dimitrios Chlorogiannis, Georgios Vasilagkos, Konstantinos Katsanos, Konstantinos Toutouzas, Adel Aminian, Dimitrios Alexopoulos, Periklis Davlouros, Grigorios Tsigkas. Safety and efficacy of shortened dual antiplatelet therapy after complex percutaneous coronary intervention: A systematic review and meta-analysis,Hellenic Journal of Cardiology,2023;71:Pages 33-41,https://doi.org/10.1016/j.hjc.2023.01.005.

Greco A, Mauro MS, Capodanno D, Angiolillo DJ. P2Y12 Inhibitor Monotherapy: Considerations for Acute and Long-Term Secondary Prevention Post-PCI. Rev Cardiovasc Med. 2022 Oct 17;23(10):348. doi: 10.31083/j. rcm2310348

4.Along the previous point, the use of CCTA to reduce antithrombotic therapy burden could be pivotal in atrial fibrillation (AF) patients with extremely high bleeding risk in whom anticoagulant and antiplatelet therapy should be needed in case of DES implantation. In this clinical scenario, the CCTA stratification could be pivotal to defer invasive management in patients with AF but without significant plaques. In this topic the authors could cite this recent paper discussing about the outcomes optimization strategies in AF patients undergoing PCI: doi:10.3390/jcdd12040142. GIVEN
Recently, in 2025, Castiello et al, reiterated the use of CCTA in reducing the burden of antithrombotic therapy that may be pivotal in patients with atrial fibrillation (AF) with extremely high risk of bleeding in whom anticoagulant and antiplatelet therapy may be required in case of implantation of drug eluting stent (DES). In this condition, the stratification of CCTA may be pivotal to alter invasive management in patients with AF but without significant plaques. Using aspirin and a P2Y12 inhibitor, as dual antiplatelet therapy (DAPT) may be necessary in all patients undergoing PCI for prevention of complications due to thrombosis. The majority of the patients undergoing PCI also have coexisting AF, thus requiring an oral anticoagulant (OAC) for prevention of stroke due to ischemia or systemic embolism. However, there is increased risk of bleeding in association of OAC and DAPT. It seems that dual antithrombotic therapy (DAT) has lesser bleeding events, compared with a triple antithrombotic therapy (TAT), that may be at the cost of higher risk of stent thrombosis. Interestingly, AF patients undergoing PCI represent a special population with many challenges, requiring multiple strategies to decrease the risk of complications due to bleeding.

Castiello DS, Buongiorno F, Manzi L, Narciso V, Forzano I, Florimonte D, Sperandeo L, Canonico ME, Avvedimento M, Paolillo R, Spinelli A, Cristiano S, Simonetti F, Semplice F, D'Alconzo D, Vallone DM, Giugliano G, Sciahbasi A, Cirillo P, Gragnano F, Calabrò P, Esposito G, Gargiulo G. Procedural and Antithrombotic Therapy Optimization in Patients with Atrial Fibrillation Undergoing Percutaneous Coronary Intervention: A Narrative Review. J Cardiovasc Dev Dis. 2025 Apr 8;12(4):142. doi: 10.3390/jcdd12040142.

5.(Ref. 30) A brief mention of the concordance between CCTA and intracoronary imaging techniques (OCT and IVUS) about the plaque morphology could be useful to have a more comprehensive vision of the diagnostic tools to describe and characterize the plaques. GIVEN
Morphology and Characteristics of Plaque.

the concordance between CCTA and intracoronary imaging techniques (OCT and IVUS) about the plaque morphology, such as coronary artery calcium (CAC) could be useful to have a more comprehensive vision of the diagnostic tools to describe and characterize the plaques [29,30]. It is possible that the combination of CAC score with a strategy of primary prevention in patients with intermediate-risk with a family history of CAD may be useful. In a randomized trial, 179 patients were included in the CAC score-informed and 186 in the usual care groups [29]. Compared with usual care control group, the CAC score-informed group revealed a sustained decline in total cholesterol levels at 3 years, in association with a significant decrease in pooled cohort equation risk calculation, P < .001). Plaque progression was higher in the control group compared with CAC group for total plaque volume; P = .009), in association with a decrease in atherogenic lipids. In the REVEALPLAQUE study [30], examined the agreement between AI-enabled quantitative coronary plaque analysis (AI-QCPA) and intravascular ultrasound (IVUS) for the assessment of plaque in the coronary artery. This study reported a strong agreement and high accuracy between AI-QCPA and IVUS for measurement of minimal luminal area, total volume of the plaque, and classified the plaque using the system of CAD-RADS. This study also found the diagnostic performance of AI-QCPA for the quantitative detection of significant stenosis. 

6.There is no mention of longitudinal data showing how plaque staging progresses over time in individual patients or how medical therapy modifies these stages. Without time-series data, the value of staging as a management tool is limited. Included: Cohort studies
Cohort studies using imaging methods such as CT angiography (CTA) and other imaging techniques can track progression of plaque and the effect of various therapies on it. These studies often categorize plaque by atheroma volume or volume, indicating for comparison over time and across various groups of treatment. In a case study involving 303 patients, definition of plaque stage thresholds of 0, 250, 750 ​mm3 and 0, 5, and 15% percent atheroma volume (PAV) indicated in 4 clinically distinct stages in which patients were distributed with no, non-obstructive, single vessel disease and multi-vessel disease [3]. It was concluded that atherosclerotic plaque burden by quantitative computed tomography (QCT) may be related to severity of stenosis and extent as well as ischemia. The authors proposed the staging of the burden of CAD atherosclerotic plaque via definitions, as given in the Table 1.

Later on, Freemen and co-workers, the American College of Cardiology Innovations in Prevention Working Group introduced, the Atherosclerosis Treatment Algorithms that personalize medical interventions based upon atherosclerosis findings from CCTA and  risk factors of CVDs [17]. It seems that via integration of CCTA-based evaluation of atherosclerosis, guideline for clinical practice, and evidence from contemporary randomized controlled trial, the treatment of Atherosclerosis Algorithms can leverage patient-specific atherosclerosis progression and burden as primary targets for interventional therapy. The stages of severity of atherosclerosis are defined by CCTA, followed by Atherosclerosis Treatment Algorithms which are described for various stages of worsening atherosclerosis for patients with obesity, diabetes mellitus, lipid disorders, hypertension, and tobacco use [17]. It is possible that there would be a rapid pace of research in the field, due to the perspectives on future needs that may improve efforts to optimize the prevention of CAD. This approach may allow the experts to focus efforts on assessing and managing risk of adverse events rather than fixing a lesion or two periodically [31-33]. This approach is consistent with a precision cardiac care which emphasizes atherosclerosis as the target of primary disease for evaluation and therapy.

According to guidelines, non-invasive imaging offers a method to identify CAD, to improve risk stratification, and to guide patient management [34,35]. Large randomized controlled trials have demonstrated the convincing data regarding use of CCTA for patients with stable CAD [36,37]. The Scottish Computed Tomography of the Heart (SCOT-HEART) recruited, 4146 patients, mean age 57 years, 2325 (56·1%) male, with 2073 randomly assigned to standard care and CCTA and 2073 to standard care alone [36,37]. This randomized controlled trial found that in patients with suspected angina due to CAD, CCTA led to a change in diagnosis in 27%, a change in investigations in 15%, and a change in treatment in 23% of patients [36].  3 Follow-up, after 5 years, revealed that this therapy led to a decrease in the composite endpoint of death due to CAD or non-fatal MI compared with standard care, with similar rates of coronary revascularisation and an increase in the use of preventive therapies [37]. 

 After a median follow up of 10·0 years, deaths due to CAD or non-fatal MI was less frequent in the CCTA group compared with the standard care group (137 [6·6%] vs 171 [8·2%]; hazard ratio [HR] 0·79, p=0·044) [38]. While all-cause, cardiovascular, and death due to CAD, and non-fatal stroke, were similar between the groups (p>0·05 for all), but non-fatal MI (90 [4·3%] vs 124 [6·0%]; HR 0·72, p=0·017) and MACE (172 [8·3%] vs 214 [10·3%]; HR 0·80, p=0·026) were less frequent in the CCTA group [38]. The therapy for prevention remained more frequent in the CCTA group (831 [55·9%] of 1486 vs 728 [49·0%] of 1485 patients with available data; odds ratio 1·17, p=0·034). It is clear, that after 10 years, CCTA-guided therapy of patients with stable chest pain was associated with a sustained reduction in deaths due to CAD. or non-fatal MI. It is clear that identification of coronary atherosclerosis by CCTA causes significant improvement in the long-term prevention of CVDs in patients with stable chest pain with similar rates of coronary revascularisation procedures (15·2% vs 15·3%, p=0·99)

In addition, protective effects of a more accurate diagnosis may occur due to modifications of diet and lifestyle, better use of preventive therapy, and increased availability of medical services. The therapeutic effects of statin therapy remained found beyond 16 years in the Anglo-Scandinavian Cardiac Outcomes Trial Legacy study [39].   5 and more than 20 years in the West of Scotland Coronary Prevention Study [40]. 6 Interestingly, CAD is a progressive process, which may attenuate the legacy benefits on outcomes in the SCOT-HEART trial over the longer term follow up. This is more so in those intervention trials, in which the standard of care includes greater prescription of therapies aimed at prevention as patients age. There is a need to ascertain if the protective effects of management guided by CCTA persist after 10 years and whether there are subgroups of patients who get maximum benefit.

7.The primary reference for the staging system (Min et al.) is based on a cohort of only 303 patients. Conclusions about population-level applicability or treatment guidance should be made cautiously, with acknowledgment of the sample size limitation. ADDED in the Conclusion.

The primary reference for the staging system is based on a cohort of only 303 patients, hence conclusions about population-level applicability of CCTA or treatment guidance should be made cautiously, with the sample size limitation.

8.The review does not critically discuss the potential confounding variables (e.g., age, sex, comorbidities) that may affect plaque burden, composition, and prognosis. Any future validation of the staging system should involve multivariate modeling to isolate independent predictive power.

Risk Factors and Protective Factors Influencing Atherosclerotic Plaque.

There is evidence that ACAD can develop over the life course, beginning from preconception period, intra-uterine life and off-spring [1,4,20] with early features of atherosclerosis evident as early as the first decade of life [4].The risk of ACAD may be increased due to multiple factors and may be broadly categorized as genetic, behavioral, biological, metabolic, environmental, or related to other comorbidities such as diabetes mellitus. It seems that several risk factors for ACAD are well recognized, however, the emphasis is now moving on novel, emerging, and undiscovered risk factors, which may require further research across diverse populations and age groups [4,20]. These risk factors of ACAD are expected to evolve considerably as diet and lifestyle change with alteration in ecology, environment, and demography with technological unfolding of advancements. These potential confounding variables such as age, sex, comorbidities may affect plaque burden, composition, and prognosis. Therefore, any future validation of the staging system should involve multivariate modeling to isolate independent predictive power.

9.The review does not address that patients selected for CCTA may already be at moderate to high cardiovascular risk, introducing potential selection bias when generalizing the utility of the staging system to broader populations.. In the manuscript it could be helpful adding a brief section discussing the clinical scenarios in which the CCTA could not be the preferred diagnostic tool to find and manage CAD.
Selection of Patients for CCTA and None Indications.

CCTA is a non-invasive imaging technique that helps the diagnosis and typically used in patients with symptoms suggestive of or on high risk of CAD. The patients selected may already be at moderate to high risk of CVDs, introducing potential selection bias when generalizing the utility of the staging system to broader populations. This means the study results might not be generalizable to the broader population due to the specific characteristics of those being tested. There is evidence that in patients with extensive CAC, CCTA had a negative predictive value (> 90%) to diagnose lack of significant stenosis, and often overestimated presence of coronary stenosis [41]. Therefore, the prevalence of CAD in patients undergoing CCTA, is often high, causing a skewed sample, which means the findings may not accurately reflect the true prevalence of CAD in the broader population. The patients that are studied are already a pre-selected group with a higher likelihood of having the condition. This potential bias is crucial for interpretation of the findings of studies via CCTA, because a high sensitivity and net percent value in a high-risk population may not translate to the same extent of accuracy in a lower-risk population. Therefore, in patients with lower risk of CAD, where false-positive results could lead to unnecessary invasive procedures, the CCTA may not be the preferred diagnostic method.

10.The integration of artificial intelligence or machine learning models to refine staging and outcome prediction could be helpful in this clinical scenario. Please, expand this section.
Artificial Intelligence

In patients with lower risk of CAD, where false-positive results could lead to unnecessary invasive procedures, the CCTA may not be the preferred diagnostic method. The integration of artificial intelligence or machine learning models to refine staging and outcome prediction could be helpful in this clinical scenario. There is a need to develop high-quality evidence to make this effort meaningful, by using artificial intelligence that has allowed for more effective automated plaque evaluation across the entire coronary tree [30], and for prediction of future CVD events [31]. In a case study, involving 351 patients, aged mean 65.9 years, the median interval from CCTA to ACS event was 375 days, and 223 patients (63.5%) had MI. In the derivation cohort (n = 243), the best AI-QCPHA features were fractional flow reserve for the identification of the burden of plaque, lesion, total volume of plaque, low-attenuation volume of plaque, and total myocardial blood flow. There was a higher predictability than the reference model in the validation cohort (n = 108) (AUC: 0.84 vs 0.78; P < 0.001) due to the addition of AI-QCPHA features. It is clear that AI-enabled plaque and hemodynamic quantification enhanced the predictability for culprit lesions in ACS patients, over the conventional coronary CTA analysis.

11.To improve the impact of this paper the authors could provide a graphical picture with CCTA, plaques and stenting images discussing the key advantages and disadvantages of the discussed staging classification of ACAD.

12.Furthermore, a structured table summarizing the strengths and limitations of the cited studies on CCTA and plaque staging. Could be really appreciable.

Please, reconcile the text’s character in all the text and use only an order of numbers in the bibliography?? Not clear.

We are sorry, that we cannot address point 11 and 12, but if the referee is kind enough to provide this support through any of his students, whose names may be included in this paper. After all, the purpose of the review and of any journal by a referee, is to improve the article for audience.

Reviewer 2 Report

Comments and Suggestions for Authors

Interesting review 

However There are some scoring systems that you didn t mentioned like GENSINI score which reflects quantitative  severety of atherosclerosis, Syntax, IVUS and others. Calcium score in my opinion is less important than others/

You must discuss it / why we need another scoring system /

You must also discuss what practical usefulness of this score

The recommended treatment based mainly on clinical picture and risk factors/

The sentences in lines 163 and 200 looks me un appropriate !!!

Please remove them (sperm and ova ..)

Author Response

Refree 2.

Interesting review. Grateful for appreciation

1.However There are some scoring systems that you did not mentioned like GENSINI score which reflects quantitative  severety of atherosclerosis, Syntax, IVUS and others. Calcium score in my opinion is less important than others/ 161-174. I agree with you, further details on sensitivity are given and highlighted, with subheading. What is Syntax?You must discuss it / why we need another scoring system /You must also discuss what practical usefulness of this score. Given below, under, Sensitivity, Specificity and Predictive Value,  highlighted

This scoring system increases the sensitivity of assessment of severity of atherosclerosis. The Gensini score is commonly used for quantifying the severity of CAD based on angiography [20]. It assigns points based on the degree of stenosis in coronary arteries, with higher scores indicating greater disease burden. Despite it is not a perfect measure, the Gensini score is considered more informative than simply classifying patients based on double, or triple or single vessel disease. 

2.The recommended treatment based mainly on clinical picture and risk factors/ Addressed-highlighted, The Use of CCTA to Reduce Antithrombotic Therapy Burden

3.The sentences in lines 163 and 200 looks me in-appropriate !!! If necessary to delete, please be kind to clarify little bit more.

4.Please remove them (sperm and ova). Explains mechanisms how fathers risk factors predispose transgenerational epigenetic risk from father to offspring, but may be deleted.

A father's diet before conception can influence the health of his offspring through transgenerational epigenetic inheritance, potentially increasing the risk of chronic diseases like metabolic disorders and cardiovascular diseases. This happens because the father's diet can alter the epigenome of sperm, which carries epigenetic modifications to the offspring, potentially affecting gene expression and development

Ng SF, Lin RC, Laybutt DR, Barres R, Owens JA, Morris MJ. Chronic high-fat diet in father’s programs β-cell dysfunction in female rat offspring. Nature. (2010) 467:963–6. doi: 10.1038/nature09491

Carone BR, Fauquier L, Habib N, Shea JM, Hart CE, Li R, et al. Paternally induced transgenerational environment reprogramming of metabolic gene expression in mammals. Cell. (2010) 143:1084–96. doi: 10.1016/j.cell.2010.12.008

Reviewer 3 Report

Comments and Suggestions for Authors

Its a good paper. I suggest to improve  the discussion about functional study to identify vulnerable plaques, and neovascularization of the atheroma as local inflammation markers.

Author Response

Refree 3.

Its a good paper. Grateful for appreciation

I suggest to improve the discussion about functional study to identify vulnerable plaques, and neovascularization of the atheroma as local inflammation markers.

Please see discussion, para 4,  highlighted

Identification of vulnerable plaques, and neovascularization of the atheroma as local inflammation markers are crucial for prevention of ACAD [11,18]. In vulnerable lesions of carotid plaques, the mRNA levels of matrix metalloproteinase (MMP)-2, -7, -9, and -14 were higher, and the protein levels of MMP-2 and -14 were also increased [18]. There was overexpression of vascular endothelial growth factor [VEGF], bone sialoprotein 2 , indicating neovascularization and calcification and the protein levels of extracellular regulated kinase (ERK) and protein kinase C (PKC), in the vulnerable plaques, pointing to the role of that MMP-2 and -14 in plaque vulnerability.

Round 2

Reviewer 1 Report

Comments and Suggestions for Authors

This is a second revision of well executed research and a well written paper aimed to emphasize the role of a staging system proposed by the Lancet Commission for quantification of the atherosclerotic coronary artery disease (ACAD), based on coronary computed tomography angiography (CCTA).

The topic is of clinical relevance. The paper is well revised.

There are no concerns about ethical issues, conflict of interest and plagiarism/publication ethics.

The paper can be accepted in its present form.

Reviewer 2 Report

Comments and Suggestions for Authors

The report in rebuttal letter are not specific and did answer on my remarks

You ask me what is SYNTAX

You must know...